# Development and Characterization of Cultured Buttermilk Fortified with *Spirulina plantensis* and Its Physico-Chemical and Functional Characteristics

Hency Rose [1], Shiva Bakshi [1], Prajasattak Kanetkar [1], Smitha J. Lukose [2], Jude Felix [1], Satya Prakash Yadav [1], Pankaj Kumar Gupta [1] and Vinod Kumar Paswan [1,*]

[1] Department of Dairy Science and Food Technology, Institute of Agricultural Sciences, Banaras Hindu University, Varanasi 221005, India
[2] Department of Dairy Chemistry, Verghese Kurien Institute of Dairy and Food Technology, Kerala Veterinary and Animal Sciences University, Mannuthy, Thrissur 680651, India
* Correspondence: vkpaswan.dsft@bhu.ac.in

**Abstract:** In recent years, there has been an unprecedented increase in the demand for fermented dairy products due to medical recommendations and lifestyle preferences. Cultured buttermilk, as an ancient fermented dairy beverage, is an appropriate product choice in this context. This study presents a novel cultured buttermilk formulated by fortification with high protein microalgae *Spirulina platensis*, thus making it valuable and attractive because of its antioxidant properties. The fermentation process, nutraceutical properties, and sensory characteristics of developed cultured buttermilk with various concentrations of *Spirulina* (0.25, 0.5, and 1%) were compared with the control sample (0% *Spirulina* buttermilk). Different concentrations of *Spirulina* in buttermilk result in a significant increase in chlorophyll and carotenoid content, boosting its antioxidant properties. The study also evaluated the prebiotic properties of *Spirulina*, thus, demonstrating its ability to promote a healthy digestive system. It was found that the addition of 0.25% *Spirulina* was able to ferment the product more quickly and retained the sensory acceptability of the finished product. The protein content, free radical scavenging activity, chlorophyll, carotenoid, and total phenolic content of 0.25% *Spirulina*-fortified buttermilk was 1.83%, 48.19%, 30.9 mg/g, 8.24 mg/g, and 4.21 mg/g GAE, respectively. Based on the results obtained, it was concluded that cultured buttermilk with a high nutritional value and functional health benefits can be developed by fortification with 0.25% *Spirulina* as a natural ingredient.

**Keywords:** cultured butter milk; dairy byproduct; functional foods; nutraceuticals; *Spirulina*; fortification

## 1. Introduction

The nutritional and health benefits of traditional fermented dairy products drive growing consumer interest because they promote the intestinal bacterial microbiota thus promising healthy living and longer life expectancy [1,2]. Fermented milk products have been an integral part of Indian diets since ancient times. The products are the result of interactive processes that occur while surplus milk is being stored [3]. However, it was Louis Pasteur who first provided an explanation for these interactive processes through "fermentation theory" [4]. Fermented milk products have enhanced bioavailability, shelf life, and safety contributed by the bioactive metabolites produced by lactic acid bacteria (LAB). Fermented milk products possess desired and predictable sensory, rheological, nutritional, and functional changes, which have increased the economic value of fermented milk products.

When fermented milk or cream is churned into butter, buttermilk is produced as a byproduct, which is classified as a functional food, predominantly due to its emulsifying

power contributed by the polar lipids and milk fat globule membrane (MFGM) components [5]. It is possible to describe buttermilk as traditional/conventional buttermilk or as cultured buttermilk [6]. Churning is used to separate the white butter from the curd, and the remaining liquid is known as cultured buttermilk. Curd manufacturing, curd homogenization, and water dilution are all steps in India's commercial manufacture of cultured buttermilk [7]. Cultured buttermilk is a nutritious and health-promoting food beverage because it is higher in phospholipid content and an excellent source of calcium, phosphorus, vitamin B2, vitamin B12, pantothenic acid-vitamin B5, potassium, zinc, protein, iodine, and molybdenum. As a result of the consumption of cultured buttermilk in the diet, one can improve digestion, increase immunity, and reduce cholesterol levels in the blood [8]. The term "buttermilk" is ambiguous and can be confused with or related to a range of other products, including natural buttermilk, cultured buttermilk, sour milk, cultured milk, cultured skimmed milk, certain Scandinavian fermented milks, and Bulgarian fermented milks, depending on the region [9].

Cultured buttermilk is a well-known fermented dairy product with medicinal benefits, sometimes referred to as "Chhash" in South Asian nations [10]. Due to its pleasant taste and remarkable health benefits, buttermilk has become more and more popular among consumers and is a crucial component in many Ayurvedic compositions [11]. Churning cultured cream or cultured milk curd is the traditional method for making cultured buttermilk. Adding culture to low-fat milk, followed by fermentation and homogenization, is the current method for making cultured buttermilk. The commercial manufacture of dahi, or cultured buttermilk, in India uses a mixed strain of thermophilic and mesophilic homofermentative bacterial cultures.

Despite the high nutritional benefits, cultured buttermilk is naturally deficient in vitamin C, iron, and dietary fiber [12]. Therefore, it is essential to examine buttermilk's qualities and supplement it. One of the best strategies to improve the overall dietary intake of food with the fewest negative consequences, according to nutritionists, is to fortify milk products using natural resources [13]. Protein-based natural additives are now replacing the use of synthetic food additives that may have neurological, gastrointestinal, and respiratory disturbances, and side effects [14,15]. Almost all natural additives come from plant derivatives and extracts. Throughout history, microalgae have been used as a natural additive. Over the past few decades, researchers have observed that algae can be used as a protein source. A variety of microalgae have been studied. However, *Spirulina*, namely *S. platensis*, has been studied more than others because of its rich components, beneficial effects, and nontoxic properties [16].

*Spirulina platensis* is a microalga that has high macro and micronutrient content that helps to fight hunger and malnutrition. *Spirulina* is a nutrient-dense food that can be used therapeutically, addressing a wide array of nutritional requirements. The Intergovernmental Institution for Micro-algae *Spirulina* Against Malnutrition (IIMSAM), an intergovernmental institution, has advocated *Spirulina* as a high-nutrient food since the mid-1970s for pregnant women and newborn infants [17]. Additionally, the National Aeronautics and Space Administration (NASA) and the European Space Agency (ESA) have both suggested *Spirulina* as the best and most sustainable space food (for long-term space missions) because of its concentrated macro- and micronutrient content [18]. *Spirulina* has demonstrated efficacy as a nutraceutical food for several ailments. As a dietary supplement, *Spirulina* has reportedly been shown to be effective in preventing and treating hypercholesterolemia, as well as managing allergies, cancer, toxicity, cardiovascular disease, and diabetes [19,20]. This is because of the chemical components of the food, which contains high levels of protein along with all nine required amino acids, vital fatty acids, minerals, pigment, and vitamins. In addition to providing nourishment, it has been discovered to have antioxidant, antibacterial, and antifungal properties against several human infections [21], and to encourage the formation of lactic acid bacteria in milk and dairy products [22]. *Spirulina platensis* has been used in a variety of products, including ayran, a fermented dairy beverage [23], Kefir [24], yoghurt [22,25–28], soy-yogurt biscuits [29], ice

cream [30–32], cheese [33,34], protein bars [35], and others. To develop high-quality foods with a greater nutritional content, microalgae have been commercially exploited and used as functional components [36]. However, there have been no reports to date regarding fortification of buttermilk with *Spirulina*.

With this background, the current study was planned to produce spiced buttermilk fortified with *Spirulina platensis* to standardize the optimal concentration of *Spirulina* as a supplement in spiced buttermilk. Further, the effect of *Spirulina* on the physico-chemical properties and antioxidative activity of the final product as well as its shelf life were assessed.

## 2. Materials and Methods

Pasteurized and standardized double-toned milk with 1.5% fat and 9.0% SNF was procured from Amul Milk outlet, Varanasi, India and stored at 4 °C until required. A freeze-dried direct vat-set (DVS) yoghurt culture (NCDC-167) containing a mixed strain of thermophilic and mesophilic homofermentative bacterial culture was obtained from the National Collection for Dairy Culture, Karnal, India. The culture was stored at −18 °C until used. *Spirulina* was procured from Heilen Biopharm Pvt. Ltd., India and was stored in a refrigerator until used. The nutritional composition, antioxidant activity, and total phenolic content of the procured *Spirulina* powder was evaluated adopting a standard protocol.

### 2.1. Preparation of Cultured Buttermilk

Preparation of probiotic buttermilk with *Lactobacillus reuteri* was optimized by Rodas et al. [37]. The same protocol was used for the development of fortified spiced buttermilk with minor modifications and by incorporation of *Spirulina* powder.

To produce the *Spirulina*-milk blend, *Spirulina* biomass (85–90%) was added to toned milk at a 2% ($w/w$) concentration at the room temperature (27 °C). The preliminary study indicated that at this rate (2% $w/w$), *Spirulina* was completely soluble. Direct addition of *Spirulina* after the production of buttermilk was found undesirable due to the insoluble particles of *Spirulina* in the sample. Therefore, a constant concentration of the *Spirulina*–milk blend was produced. Afterward, to ensure optimal mixing of the *Spirulina* and milk, the mixture was homogenized using a homogenizer at pressures of 100 kg/cm$^2$ for the first stage and 200 kg/cm$^2$ for the second stage. To obtain aliquots of 0%, 0.25%, 0.50%, and 1.00% ($w/w$) *Spirulina* biomass, this mixture was diluted using toned milk. Spices (0.4% $w/w$) were added to the control milk (0%) and the *Spirulina*-milk blend followed by pasteurization at 90 °C for 10 min in a water bath. The pasteurized samples (100 mL each) were put into polystyrene cups that were pre-sterilized and were allowed to cool to 4 °C. These aliquots received an inoculation of 0.5% ($v/w$) NCDC-167 stock cultures. All of the samples were carefully wrapped in aluminum foil, stored in low density polyethylene (LDPE) bags, and incubated for seven hours at 42 °C in a temperature-controlled incubator. The samples were observed hourly until they attained a pH of 4.6–4.7 and an acidity of 0.8–0.9%. Curd setting was observed after 7 h of complete fermentation. Then, agitation or breaking of curd was performed using a laboratory blender at a speed of 10,000 rpm for 90 s. As soon as the cultured buttermilk samples were prepared, they were stored under refrigeration at 6–8 °C. The formulated cultured buttermilk samples were referred to as; $T_0$ (Control Buttermilk), $T_1$ (0.25% *Spirulina* Buttermilk), $T_2$ (0.5% *Spirulina* Buttermilk), and $T_3$ (1% *Spirulina* Buttermilk). Post development, these samples were further analyzed adopting standard scientific methodology after refrigeration for 7, 14, 21, and 28 days.

### 2.2. Physico-Chemical Analysis

After calibrating the pH meter with calibrated pH buffer solutions (4.0, 7.0, and 10.0), the pH of the unfortified and *Spirulina*-fortified cultured buttermilk samples was determined. Titratable Acidity (TA) was determined and calculated as a percentage of the lactic acid that was produced. The total solids of samples were determined using an oven in which the dry samples were kept at 105 °C overnight till a constant weight was

attained. Dry samples were ignited in a muffle furnace at 550 °C to burn all the organic components and measure the amount of ash present. The protein level was determined by the Kjeldhal method using a nitrogen conversion factor of 6.38. The Gerber method was used to determine the sample's fat content [38]. For determination of calcium, the method was adopted from Sehgal [39]. The total carbohydrates and energy (Kcal/100 mL) in the sample were calculated using the procedure of Cunniff [40]. All the analyses were performed in triplicate.

*2.3. Antioxidant Analysis*

2.3.1. DPPH Inhibition Activity

The method described by Kang and Saltveit [41] was used to measure the free radical scavenging activity of samples using the indicator 1,1-diphenyl-2-picrylhydrazil (DPPH). The DPPH radical scavenging activity was evaluated at 517 nm, and the results were presented as percentage of inhibition. All reactions were carried out in triplicates.

2.3.2. Analysis of Total Phenolic Content

The total phenolic content was determined by the Folin–Ciocalteu method [42] with some modifications. A total of 0.5 mL of the sample extract was taken in a test tube and 2.5 mL of diluted Folin–Ciocalteu (FC) Reagent was added followed by 1 mL of 7.5% $Na_2CO_3$. The test tube was covered with aluminum foil and incubated in the dark at room temperature for 1 h. Absorbance was measured at 760 nm in UV-1800 spectrophotometer. Gallic acid (0–800 mg/L) was used to produce standard calibration curve. The total phenolic content was expressed in mg of Gallic acid equivalents (GAE)/100 g. The standard curve with regression equation (y = 0.0769x + 0.0041, $R^2$ = 0.998) was obtained using the absorbance against the Gallic acid concentration (mg/100 mg).

2.3.3. Analysis of Pigments

The chlorophyll and carotenoid content in the samples was determined as per Kumar et al. [43] and Lichtenthaler and Wellburn [44].

A total of 1 mL of each sample was centrifuged at 5000 rpm for 10 min. The pellet was dissolved in 1 mL of ethanol and then sonicated at 65 °C for 30 min. After sonication, the solutions were centrifuged at 10,000 rpm for 5 min. The pigment content was estimated by measuring the supernatant absorbance (A) at 666, 653, and 470 nm and calculated using the following equations:

$$\text{Chlorophyll}_a \left( \frac{\text{mg}}{\text{L}} \right) = 15.65 \times A_{666} - 7.340 \times A_{653} \tag{1}$$

$$\text{Chlorophyll}_b \left( \frac{\text{mg}}{\text{L}} \right) = 27.05 \times A_{653} - 11.21 \times A_{666} \tag{2}$$

$$\text{Total Chlorophylls} \left( \frac{\text{mg}}{\text{L}} \right) = \text{Chlorophyll}_a + \text{Chlorophyll}_b \tag{3}$$

$$\text{Carotenoids} \left( \frac{\text{mg}}{\text{L}} \right) = (1000 \times A_{470} - 2.860 \times [\text{Chlorophyll}_a] - 85.9 \times [\text{Chlorophyll}_b])/245 \tag{4}$$

Determination of C-Phycocyanin content was executed as per Pan-utai and Iamtham [45]. A total of 1 mL of each sample was centrifuged at 5000 rpm for 10 min. The pellet was dissolved in 1 mL of distilled water and then sonicated for 30 min. The supernatant was collected by centrifugation at 10,000 rpm for 5 min and measured for optical density at 615 and 652 nm. C-Phycocyanin content was calculated using the following equation. For purity, a ratio of $A_{620}/A_{280}$ was used.

$$\text{C} - \text{phycocyanin} \left( \frac{\text{mg}}{\text{mL}} \right) = (OD_{615} - 0.474 \times OD_{652})/5.34$$

## 2.4. Sensory Evaluation

Sensory evaluation of the control and *Spirulina*-fortified cultured buttermilk samples was carried out using a 9-point hedonic scale of Like extremely (score-9), Like very much (score-8), Like moderately (score-7), Like slightly (score-6), Neither like nor dislike (Score-5), Dislike slightly (score-4), Dislike moderately (score-3), Dislike very much (score-2), Dislike extremely (Score-1). Twenty trained panelists were selected based on their previous experience and knowledge of sensory evaluation of dairy and milk-based products particularly those who were regular consumers of cultured buttermilk. Color and appearance, body and mouthfeel, flavor, and overall acceptability were the sensory parameters evaluated by the sensory panelists. All cultured buttermilk samples were taken out of the refrigerator and the serving temperature for samples ranged from 10 to 12 °C. Each cultured buttermilk sample was presented in a 100 g plastic cup filled with 50 g cultured buttermilk sample and labeled with a 3-digit code. Order of presentation of samples was randomized. Evaluation was divided into 3 sections: visual characteristics (color and appearance), texture (body and mouthfeel), and flavor evaluations. For visual attributes, the surface of each sample was examined in terms of any variation in color or unevenness. The visual analysis of cultured buttermilk was followed by texture and flavor evaluations.

## 2.5. Enumeration of Microorganisms

The samples comprising 1 mL of each were aseptically removed and serially diluted with 0.9% normal saline. For the enumeration of *Streptococcus thermophilus* and *Lactobacillus bulgaricus* M-17 agar and De Man, Rogosa and Sharpe (MRS) agar were used, respectively. The plates were incubated for 48 h at 37 °C [46]. Coliforms, yeast, and mold count were performed using the acidified Potato Dextrose Agar (PDA) and Violet Red Bile Agar (VRB), respectively, in accordance with the Standard Procedures for the Analysis of Milk Products [47]. The data were expressed as log CFU/mL. During the storage period, a microbial study was conducted using the spread plate method on the 1st, 3rd, 6th, 9th, and 12th days.

## 2.6. Statistical Analysis

All analytical determinations were performed at least in triplicate and values were expressed as the mean $\pm$ standard deviation (SD). One-way ANOVA and Duncan's multiple comparison tests were used to compare results with significant differences ($p < 0.05$). IBM SPSS statistics software version 19 (IBM Corp., Armonk, NY, USA) was used to perform all statistical analyses.

## 3. Results and Discussion

### 3.1. Composition of Spirulina Powder

The Spirulina powder used in the study was procured from Heilen Biopharm Pvt. Ltd., Gandhinagar, India. The nutritional composition and antioxidant activities of the Spirulina powder is presented in Table 1. The procured Spirulina powder has 5.12% moisture, 59.27% protein, and 7.02% total ash content. The protein content and ash content of Spirulina as reported by earlier researchers ranged from 53 to 65% and 7.9–8.4%, respectively, when expressed on dry matter basis [48–50]. Thus, the protein and total ash contents of the procured Spirulina powder in the present study was within the range of reported values. Further, it was reported that Spirulina extract was able to inhibit 42–50% of free radicals, containing Total Phenolic Content (TPC) 4.51–26.64 mg GAE/g samples on dry matter basis [48,51]. The variable % DPPH inhibition (89.00 $\pm$ 1.02%) and total phenolic content (11.26 $\pm$ 0.04 mg GAE/100 g) in the procured Spirulina microalga may be due to different stage of maturity and harvest and processing such as grinding and drying conditions.

**Table 1.** Nutritional composition, antioxidant activity, and total phenolic content of the procured Spirulina powder.

| Components | Value (%) |
|---|---|
| Moisture | $5.12 \pm 0.03$ |
| Protein | $59.27 \pm 0.02$ |
| Total solids | $94.88 \pm 0.03$ |
| Ash contents | $7.02 \pm 0.03$ |
| % DPPH inhibition | $89.00 \pm 1.02$ |
| Total Phenolic Content (TPC) | $11.26 \pm 0.04$ mg GAE/100 g |

Results are expressed as the mean $\pm$ standard deviation (n = 3).

### 3.2. Preparation of Cultured Buttermilk

Figure 1 shows the pH and acidity levels of the different treatments measured over the course of the fermentation (4 hours). During the development of cultured buttermilk, caseinate particles entirely precipitate over a pH range of 4.6–4.7 [52]. Therefore, the duration of fermentation in the current study was determined by the time it took for the pH to reach 4.6–4.7. Due to the strong buffering properties of milk and the rapid bacterial growth during fermentation, increasing rates of titratable acidity and decreasing rates of pH were observed across all treatments [22]. The pH of the *Spirulina*-fortified buttermilk rapidly drops after 2 h, whereas the $T_0$ takes a substantially longer period of 3 h. For instance, after 1–2 h of fermentation, the $T_3$ sample showed a pH drop from 6.42 to 4.29 (a 2.27 reduction), whereas the pH of $T_0$ declined from 6.42 to 6.2. *Spirulina platensis* was added to buttermilk samples, and as a result, the time it took to achieve pH 4.5 was considerably shortened ($p < 0.05$). The samples containing 0.25, 0.5, and 1% *Spirulina* powder did not significantly differ from one another, though. Additionally, these *Spirulina*-fortified treatments had a much faster increase in acidity ($p < 0.05$). It was found that there was a moderate agreement between the results of this study and those of earlier workers [22,25,53–56]. *Spirulina*-inoculated milk with *S. thermophilus* and *L. bulgaricus* decreased pH levels more rapidly than control samples during fermentation, resulting in a shorter incubation period.

As shown in Figure 2, each product was evaluated for color, flavor, mouthfeel, and overall acceptability. According to the hedonic scale, 82% of the trained sensory panelist indicated a positive response to $T_0$ and 78% to $T_1$. The majority of panelists underrated $T_3$ with a score of 5 on the hedonic scale. Supplemental *Spirulina* creates compounds from the oxidation of minerals and lipids that not only function as pro-oxidant molecules but may also produce metallic off-flavors, resulting in an unfavorable flavor [57]. The sample $T_1$ was not significantly different from $T_0$ (Figure 2). When panelists were exposed to buttermilk with a 1% microalgae concentration ($T_3$), it was found to have an unsuitable sensory property ($p = 0.001$). In addition, insoluble *Spirulina* particles produced a graininess mostly in $T_2$ and $T_3$, and a significant change in oral texture was observed ($p < 0.001$). Thus, it showed that the higher concentration of *Spirulina* decreases the solubility as compared to other samples, therefore higher concentrations were not accepted. The samples of $T_1$ scored the highest for taste, texture, and mouthfeel control, while treatments containing $T_2$ and $T_3$ scored the lowest ($p < 0.001$). Thus, the highest sensory score and overall acceptability was obtained for $T_1$ with 0.25% *Spirulina*-fortified buttermilk. Therefore, the buttermilk supplemented with 0.25% of *Spirulina* ($T_1$) was selected as the best final product and was evaluated for further analysis.

### 3.3. Physicochemical Analysis

Buttermilk that was enriched with *Spirulina* ($T_1$) and the control sample ($T_0$) had comparable chemical compositions for most of the attributes. Nonetheless, *Spirulina*-treated buttermilk was shown to have more total solids (TS), protein, fat, and ash concentrations (Table 2). Both the control and the *Spirulina*-fortified buttermilk had total solids of 9.73% and 9.87%, respectively. Between these two samples, there was a significant difference in TS ($p < 0.001$). The fat content of the control (0.70) and fortified buttermilk sample

(0.73) did not differ significantly ($p < 0.05$). However, the *Spirulina*-fortified buttermilk had a higher total protein value of 1.83 compared to the control sample (1.71) ($p < 0.001$). This was most likely caused by the high protein content of *Spirulina* [58]. The sample treated with *Spirulina* had the highest calcium concentration (82.05 mg/100 g on dry weight basis) ($p < 0.001$). This finding is in line with the findings of Ustun-Aytekin et al. [59], who reported that the calcium content of milk appeared to fall noticeably after fermentation, whereas *Spirulina*-enriched kefir was found to have higher levels of calcium than that of traditional kefir. Due to this high content of calcium, $T_1$ had the higher ash content. The ash content found in $T_1$ was 0.76%, which was significantly higher than the ash percentage of $T_0$ (0.70%) ($p < 0.05$). When compared to $T_0$, the energy value (Kcal/100 mL) reveals a significant difference ($p < 0.001$), with the highest value for $T_1$ (40.29). This is due to the presence of the high carbohydrate content present in $T_1$ (6.59). Changes in these factors, particularly in the amounts of protein, fat, and dietary fiber, may have an impact on other physicochemical characteristics including pH and titratable acidity.

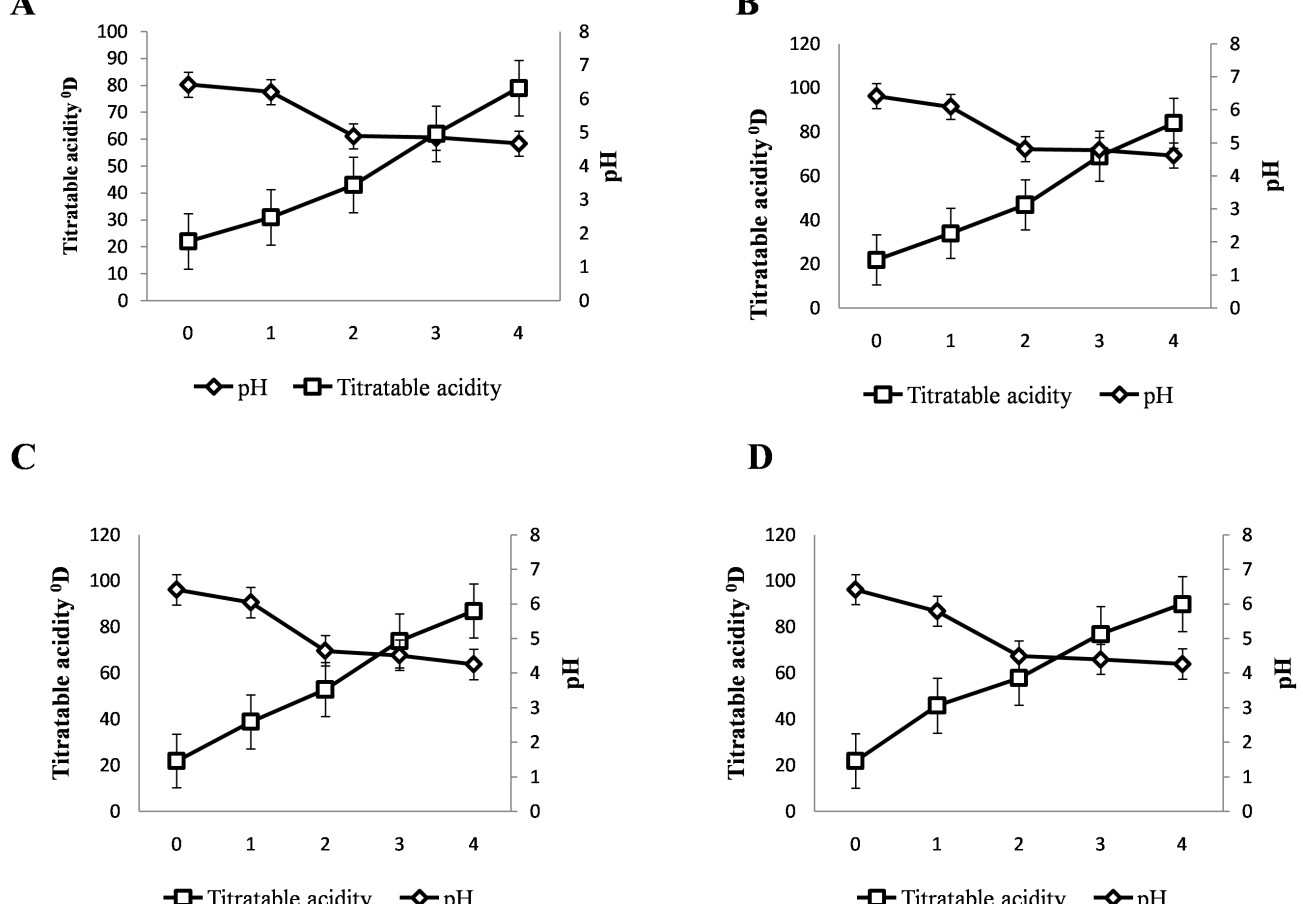

**Figure 1.** Changes in titratable acidity and pH drop in $T_0$ (**A**), $T_1$ (**B**), $T_2$ (**C**), and $T_3$ (**D**) during fermentation period (4 h).

The viscosity of a fluid describes its resistance to flow. It contributes directly to the mouthfeel characteristic of cultured buttermilk up to a certain degree. The presence of high viscosity in cultured buttermilk can also cause low phase separation since it reduces the likelihood of whey separation from the network when the viscosity of the buttermilk is high. The binding abilities or stabilizing between the serum and aqueous phase of *Spirulina* powder may have contributed to the higher viscosity (21.9 cP) values exhibited even in low concentrations of *Spirulina* powder added into cultured buttermilk. These results are similar with the findings of Mudgil et al. [7].

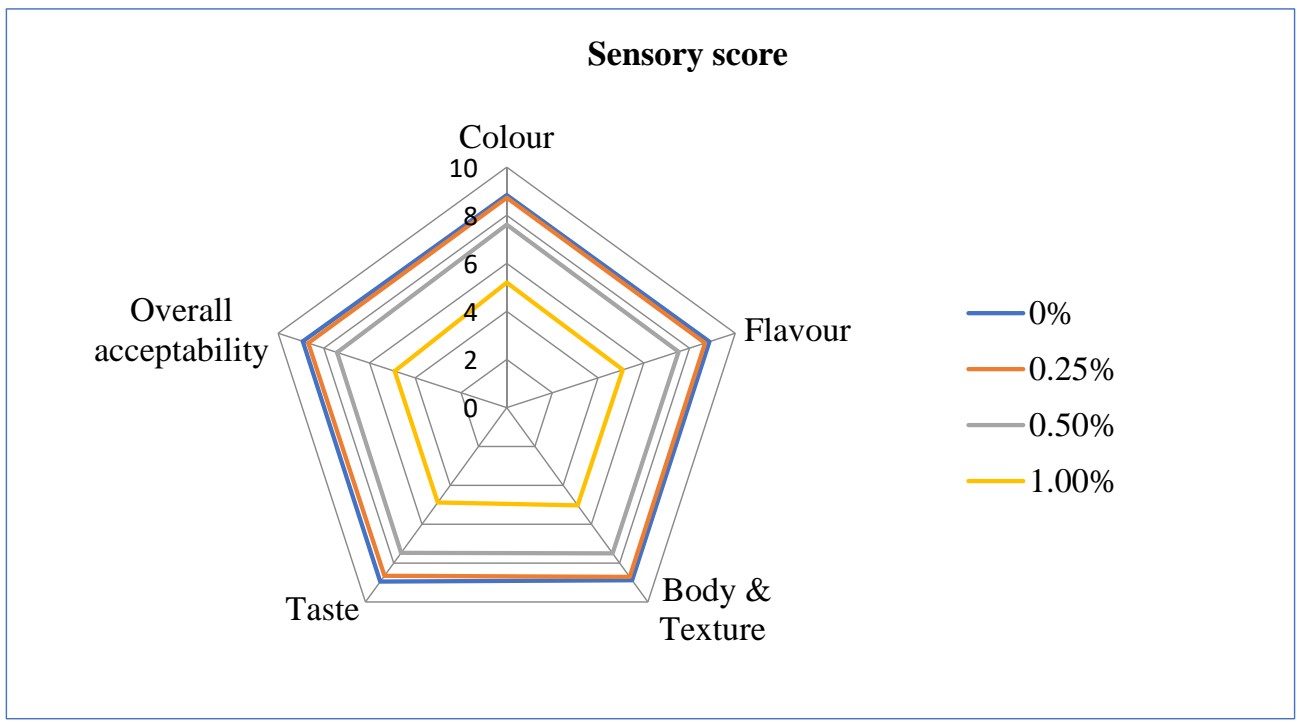

**Figure 2.** Sensory evaluation of control and different spirulina-treated samples in hedonic scale.

**Table 2.** Physicochemical Characteristics of buttermilk samples.

| Parameters | $T_0$ | $T_1$ | *p* Value |
|---|---|---|---|
| Total Solids (g/100 g on dry weight basis) | $9.73 \pm 0.01$ | $9.87 \pm 0.01$ [b] | 0.004 |
| Protein (g/100 g on dry weight basis) | $1.71 \pm 0.02$ | $1.83 \pm 0.01$ [b] | 0.003 |
| Fat (g/100 g on dry weight basis) | $0.70 \pm 0.01$ | $0.733 \pm 0.02$ | 0.065 |
| Carbohydrate (g/100 g on dry weight basis) | $6.56 \pm 0.01$ | $6.59 \pm 0.02$ [b] | 0.001 |
| Energy (Kcal/100 mL) | $39.45 \pm 0.18$ | $40.29 \pm 0.11$ [b] | 0.007 |
| Ash (g/100 g on dry weight basis) | $0.70 \pm 0.02$ | $0.76 \pm 0.03$ [a] | 0.042 |
| Calcium (mg/100 g on dry weight basis) | $74.59 \pm 0.97$ | $82.06 \pm 1.02$ [b] | 0.001 |
| Viscosity (cP) | $20.2 \pm 0.01$ | $21.9 \pm 0.01$ [b] | 0.008 |

Results are expressed as the mean $\pm$ standard deviation (n = 3). The values followed by superscript letters ([a] and [b]) are significantly different between treatment means at [a] $p < 0.05$; [b] $p < 0.01$. Control ($T_0$) vs. 0.25% Spirulina-fortified buttermilk ($T_1$).

The pH and titratable acidity of $T_0$ and $T_1$ samples during the period of 12 days of storage at 4 °C are shown in Figure 2. In the current investigation, the pH of the treated sample dropped (from 4.60 to 4.51) during storage, indicating that lactic acidity (0.22 to 0.35%) developed throughout the storage period. This finding is in agreement with the data of earlier researchers [25,60,61]; they reported that *Spirulina*-added yoghurt had lower pH values than control samples during the storage period.

### 3.4. Antioxidant Properties

A great deal of recent attention has been paid to the potential role of enriched foods that contain bioactive compounds, generally obtained from microalgae, in the treatment of many human diseases. Microalgae are content with high amounts of free radical scavengers, which explains their high antioxidant activity. Hence, chlorophyll, carotenoid contents, total phenolic content, and DPPH-radical scavenging activity were measured (Table 3) to evaluate its antioxidant capacity.

**Table 3.** Antioxidant parameters of control and *Spirulina* incorporated buttermilk.

| Parameters | $T_0$ | $T_1$ | $p$ Value |
|---|---|---|---|
| DPPH–scavenging activity (%) | $41.99 \pm 0.81$ | $48.19 \pm 0.26$ [a] | <0.001 |
| Chlorophyll (mg/g on dry weight basis) | $0.30 \pm 0.02$ | $30.91 \pm 0.04$ [a] | <0.001 |
| Carotenoid (mg/g on dry weight basis) | 0 | $8.24 \pm 0.02$ [a] | <0.001 |
| C-Phycocyanin (mg/g on dry weight basis) | 0 | $0.30 \pm 0.01$ [a] | <0.001 |
| Total Phenolic Content (mg/g GAE) | $2.44 \pm 0.01$ | $4.21 \pm 0.03$ [a] | <0.001 |

Results are expressed as the mean $\pm$ standard deviation (n = 3). Values with superscript letters are significantly different at $p < 0.001$.

### 3.4.1. DPPH Radical Scavenging Activity

As shown in Table 3, the experimental samples exhibited high antioxidant properties in terms of DPPH radical scavenging activity. The treatment sample $T_1$ exhibited significantly higher (48.19% versus 41.99%) ($p < 0.001$) radical scavenging activity. It has been reported that higher antioxidant activity is observed when a higher ratio of microalga is used [62]. Studies with yogurts fortified with *Chlorella vulgaris* and *Dunaliella* sp. have also shown the microalgae-enhanced antioxidant activity of the yogurt. Incorporating *Spirulina* powders into the diet may increase the levels of chlorophylls [63], carotenoids [64], and phycocyanin [65], resulting in an increase in free radical scavenging.

### 3.4.2. Chlorophyll Content

*Spirulina* incorporation significantly improved the chlorophyll content from 0.33 mg/g in $T_0$ to 30.95 mg/g in $T_1$ ($p < 0.001$) besides the enhancement of DPPH inhibition activities ($p < 0.05$). This result was found to be similar to findings of Barkallah et al. [25]; they investigated the effects of *Spirulina*-fortified yoghurt and reported that the chlorophyll content of *Spirulina*-fortified yoghurt was 27.6 mg/g on a dry weight basis as compared to 0.25% in the control yoghurt. According to Ismaiel et al. [63], an increase in free radical scavenging can be attributed to an increase in chlorophyll content. Marzorati et al. described a study of chlorophyll extraction methods using supercritical $CO_2$ and water [64]. They reported that the total chlorophyll content present in dried *Spirulina* powder was 9.1 mg/g. In the present investigation, the value obtained was much higher and it is inferred that the yield of chlorophyll content may vary depending on the different extraction processes.

### 3.4.3. Carotenoid Content

Carotenoids are a class of pigments found in nature. *Spirulina* contains up to 4000 mg/kg of carotenoids, with β-carotene dominating [65]. Carotenoids contain antioxidant, anti-aging, and anti-inflammatory properties, among other things. Carotenoids can quench singlet oxygen and scavenge radicals, resulting in the termination of oxidation chain reactions [66]. Because of their antioxidant characteristics, carotenoids are commonly utilized to treat cardiovascular diseases [67]. Barkallah et al. reported that the carotenoid content in $T_1$ was 10.86 mg/g on a dry weight basis in their study [25]. Szmejda et al. [68] analyzed antioxidant compounds in ice cream supplemented with *Spirulina* extract. There was a significant difference ($p < 0.001$) in the carotenoid level present in the control ($T_0$) and *Spirulina* extract-supplemented ice cream ($T_1$) which were 0.072 mg/g and 8.24 mg/g, respectively. The study of Szmejda et al. indicated that when compared to non-supplemented flavor variants, the samples enhanced with *Spirulina* extract have a 3–3.5 times larger magnitude of carotenoids. Marzorati et al. studied the supercritical $CO_2$ and water extraction methods for carotenoids, and they reported that total carotenoid content present in dried *Spirulina* powder was equal to 3.5 mg/g [66]. The yield of carotenoid content may also vary depending on the different extraction processes. The carotenoid content present in *Spirulina*-fortified buttermilk acts as a health-benefiting beverage at considerably low cost. Carotenoid as a color pigment also contributes to the antioxidant compound. However, the pigment quality present in buttermilk decreases upon storage.

### 3.4.4. C Phycocyanin Content

The C-Phycocyanin level was observed in $T_1$ with 0.028 mg/g showing a significant difference ($p < 0.001$). This compound was absent in the control buttermilk. A similar result was observed in the study of Barkallah et al. on the effect of *Spirulina*-fortified yoghurt. They reported that the C-Phycocyanin content in $T_1$ was 0.297 mg/g on a dry weight basis [25]. Marzorati et al. reported the yield of C-Phycocyanin using the water extraction method including electrocoagulation, dialysis, and protein salting-out method yielded- 250 mg/g of C-Phycocyanin with a very high purity of 2.2 [66]. Khandual et al. described different methods for extracting phycocyanin from *Spirulina* biomass (dry, wet, and frozen) [69]. They reported a greater phycocyanin content with water as the solvent, 46.65–54.65 mg/g biomass. The phycocyanin concentration of the phosphate buffer at pH 6 was 35.54–37.88 mg/g, while that of the phosphate buffer at pH 7 was 43.13–45.02 mg/g. When we compared different procedures, ultrasonication produced the highest phycocyanin content, 54.65 mg/g with water as the solvent, and the lowest, 35.54 mg/g with phosphate buffer at pH 6. It was observed that ultrasonication retrieved the most phycocyanin, followed by freeze/thaw, and sonication.

### Purity of C-Phycocyanin

C-Phycocyanin with a purity of 0.7 is regarded as food grade and between 0.7 to 3.9 is called reactive grade, and greater than 3.9 is termed analytical grade [70]. However, the purity of the experimental sample $T_1$ was 1.32%, which is above 0.7%. This shows that our results were included within the food grade range.

### 3.4.5. Total Phenolic Content (TPC)

A significant difference was observed between $T_0$ and $T_1$ for TPC ($p < 0.001$). As a result of the addition of *Spirulina platensis* to the cultured buttermilk, the total phenolic content was greatly increased in $T_1$. A similar response was observed when the microalgae was added to gluten-free pasta samples [64]. In addition, Niccolai et al. identified an increase in phenol content as a result of fermentation of *Spirulina* biomass [71]. Previously, Liu et al. reported that *Spirulina platensis* is high in phenolic content (19.47 mg/g GAE) and fermentation enhances its effects [72].

### 3.5. Enumeration of Microorganism

The changes observed in the cultures of *L. bulgaricus* and *S. thermophilus* throughout the period of 15 days of storage at 4 °C are shown in Figure 3. After the fermentation process, the *L. bulgaricus* and *S. thermophilus* counts of the buttermilk samples were analyzed, and the initial counts of the microorganisms were found to be 9.12 log CFU/g and 9.13 log CFU/g, respectively. The results showed that *Spirulina platensis* had a significant impact on the lactobacilli and streptococci counts of treated samples compared to control sample milk throughout 12 days of cold storage ($p < 0.05$). Comparable results were reported by Güldas and Irkin [73] and Sudheendra [74]. Similarly, Beheshtipour et al. [22] found that lactic acid bacteria were more likely to survive in the presence of *Spirulina platensis*. According to Alizadeh Khaledabad et al. [60], probiotic populations of yoghurt samples were boosted by an increase in *Spirulina platensis* during its shelf-life. Furthermore, *Spirulina platensis* is known to contain further growth-inducing components, such as adenine, free amino acids, and linoleic acid [75–77]. This indicates that *Spirulina* has some prebiotic efficiencies that could help the growth of the culture. Varga and co-workers [55] were the first to identify this symbiotic relationship between *Spirulina platensis* and lactobacilli. Additionally, Gupta et al. [77] investigated how blue green algae affected probiotic bacteria. Due to the prebiotic properties of *Spirulina*, the product in our study may be regarded as a symbiotic product, although further evaluations and studies need to be conducted to determine *Spirulina*'s prebiotic efficiency.

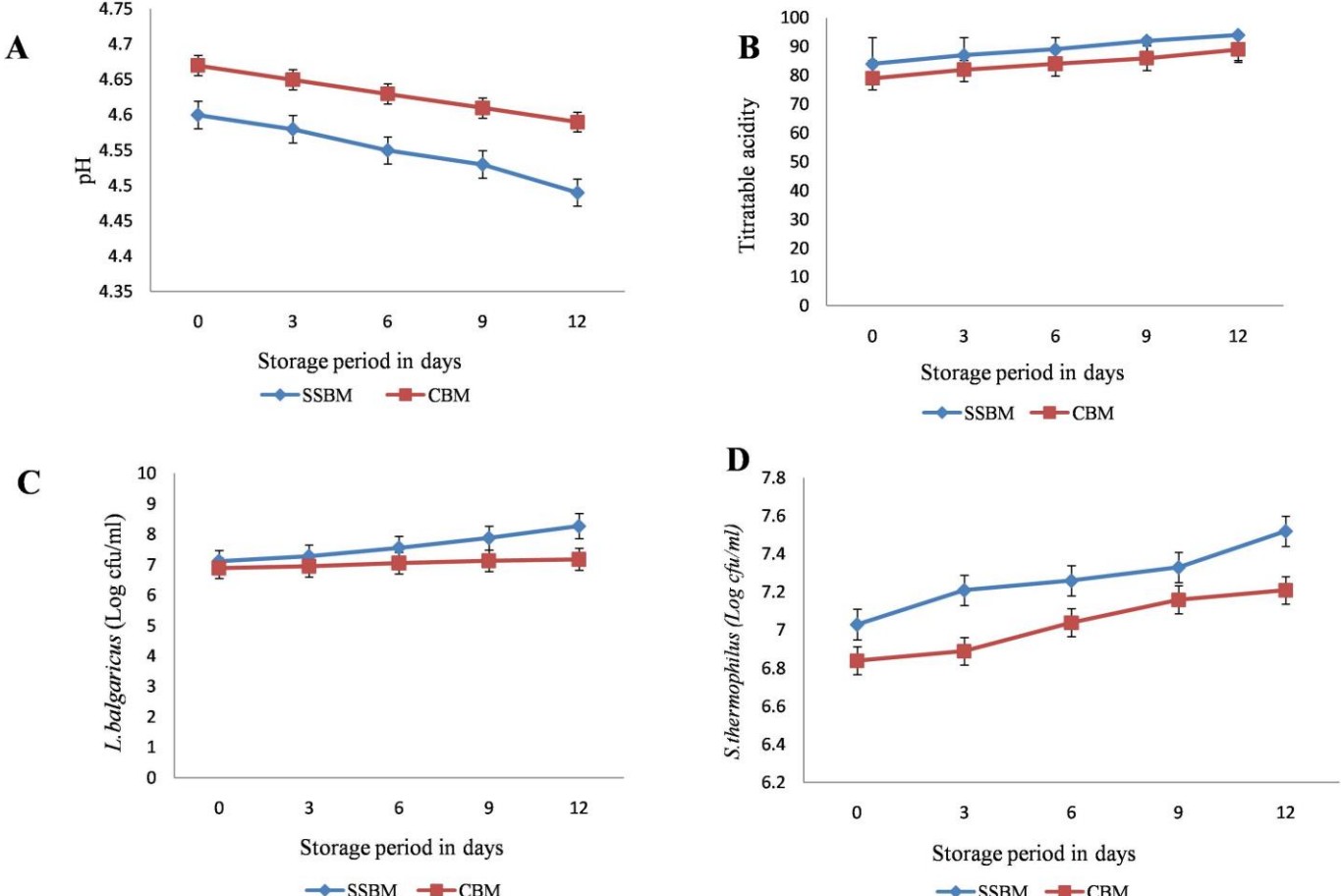

**Figure 3.** The pH (**A**) and acidity (**B**) value, *L. bulgaricus* (**C**) and *S. thermophilus* (**D**) counts (log cfumL$^{-1}$) expressed in (mean $\pm$ SD) of samples during 21 days of storage at 4 °C.

None of the samples had any evidence of mold, yeast, or coliform bacteria during storage. Even after 12 days of storage at 4 °C, the lack of these bacteria demonstrated that the buttermilk were secure and uncontaminated. These findings imply that buttermilk processing took place in clean and hygienic conditions.

## 4. Conclusions

As an innovative and attractive ingredient, *Spirulina* powder can be successfully incorporated into spiced buttermilk as a means of enhancing the nutritional and physico-chemical properties without significantly altering its sensory acceptability. The study indicated that fortifying buttermilk with 0.25% *Spirulina* enhanced nutritional and functional properties by enhancing its antioxidant activity. The microbial evidence of this investigation suggest that *Spirulina platensis* may have the ability to encourage the growth of lactic acid bacteria and can be the effective prebiotic algal source. Thus, it can be used as a high-quality dietary supplement to enhance the nutritive and bioactive properties of food and dairy products. Further research is required to overcome its unpleasant flavor by modern encapsulating techniques and the efficient extraction of bioactive components such carotenoids and C-Phycocyanin using improved extraction techniques.

**Author Contributions:** Conceptualization, H.R. and V.K.P.; methodology, H.R., P.K. and J.F.; software, S.J.L.; validation, S.B., S.P.Y. and P.K.G.; formal analysis, H.R. investigation, J.F.; resources, V.K.P.; data curation, H.R., V.K.P. and S.J.L.; writing—original draft preparation, H.R., S.P.Y. and S.B.; writing—review and editing, P.K.G.; visualization, P.K.; supervision, V.K.P. and S.J.L. All authors have read and agreed to the published version of the manuscript.

**Funding:** This research received no external funding.

**Institutional Review Board Statement:** Not applicable.

**Data Availability Statement:** The original contributions presented in the study are included in the article; further inquiries can be directed to the corresponding author/s.

**Acknowledgments:** The corresponding author acknowledges the IoE Scheme research grant provided by Banaras Hindu University, Varanasi, India.

**Conflicts of Interest:** The authors declare no conflict of interest.

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
