# Peer review of "Development and Characterization of Cultured Buttermilk Fortified with Spirulina plantensis and Its Physico-Chemical and Functional Characteristics"

_2624-862X, doi:10.3390/dairy4020019_

Round 1

Reviewer 1 Report

This study investigated the effect of microalgae fortification on fermentation process, nutraceutical properties, and sensory characteristics of cultured butter milk. The experiment design, result and writing are good. Some minor revisions are needed as follows.

Title: Not specific about what was done in this study.

Line 22-23: For which level of fortification?

Line 26: Which level of fortification is recommended?

Introduction: Need to state why Spirulina platensis was used, in comparison with other potential ingredient.

Line 223-225: More information about Spirulina should be given, especially those related to what was measured in cultured buttermilk.

Line 226: Information about butter milk?

Figure: Not visually clear.

Figure and table: Add statistical analysis

Author Response

We are grateful to the reviewer for suggesting the revisions to improve our manuscript. The reply to specific comments is presented hereunder:

Point 1: Title: Not specific about what was done in this study.

Response 1: The title has been revised to “Development and Characterization of cultured buttermilk fortified with Spirulina plantensis and its physico-chemical and functional characteristics”

Point 2: Line 22-23: For which level of fortification?

Response 2: The level of fortification has been mentioned in the revised manuscript.

Point 3: Line 26: Which level of fortification is recommended?

Response 3: The level of fortification has been mentioned in the revised manuscript.

Point 4: Introduction: Need to state why Spirulina platensis was used, in comparison with other potential ingredient.

Response 4: The justification for using Spirulina platensis have been included in the introduction part in the revised manuscript.

Point 5: Line 223-225: More information about Spirulina should be given, especially those related to what was measured in cultured buttermilk.

Response 5: As desired, we have included more information about Spirulina powder used in the study in subsection 3.1 of the results and discussion section of the revised manuscript. This has been done by inserting new Table 1 and describing and discussing its contents in subsection 3.1.

Point 6: Line 226: Information about butter milk?

Response 6: Thanks for the suggestion. However, detailed information about the formulation of cultured buttermilk has already been presented in the material and method section.

Point 7: Figure: Not visually clear.

Response 7: Earlier versions of images have been replaced by images with improved resolutions in the revised manuscript.

Point 8: Figure and table: Add statistical analysis

Response 8: As indicated, statistical analysis has been added.

Reviewer 2 Report

This is an interesting study “Development and characterization of functional cultured buttermilk fortified with Microalgae - Spirulina plantensis and the authors have done a good experimental design. However, here are some comments:

1.    Introduction section is well written.

2.    Objective of this study should be more attractive.

3.    Table 1 should present physicochemical properties of other treatments.

4.    Figures 1 and 3 are not clear please improving the resolution of these two figs.

5.    Discussion section is clear with new insights.

6.    Conclusion needs to be elaborated.

Author Response

Point 1: Introduction section is well written.

Response 1: Thanks for the encouraging comments.

Point 2: Objective of this study should be more attractive.

Response 2: We have improved the presentation of the objective of the study in the revised manuscript.

Point 3: Table 1 should present physicochemical properties of other treatments.

Response 3: Thank you very much for the valuable suggestion. However, as per our present research plan, we proceeded with our further studies only with control (T0 ) and T1.

Point 4: Figures 1 and 3 are not clear please improving the resolution of these two figs.

Response 4: The resolutions of the images have been improved in the revised script.

Point 5: Discussion section is clear with new insights.

Response 5: Thank you so much for the encouraging comment.

Point 6: Conclusion needs to be elaborated.

Response 6: The conclusion part has been elaborated as suggested in the revised script.

Reviewer 3 Report

The manuscript is clearly described. In the Materials and Methods section, describe in detail on what basis you chose to aliquot the spirulina and add a concentration of 2% (w/w) at room temperature (27°C)?

A further method of assessing antioxidant activity expressed as a concentration and not as a percentage of inhibition should be included.

In addition, another control with Spirulina alone should be included to check its antioxidant activity individually.

Review the bibliography by inserting DOIs

Author Response

Point 1: The manuscript is clearly described. In the Materials and Methods section, describe in detail on what basis you chose to aliquot the spirulina and add a concentration of 2% (w/w) at room temperature (27°C)?

Response 1: The concentrations of 2% (w/w) for Spirulina was chosen on the basis of preliminary trial (we have already mentioned this in our original manuscript) based on its dissolution at room temperature.

Point 2: A further method of assessing antioxidant activity expressed as a concentration and not as a percentage of inhibition should be included.

Response 2: Thanks for your valuable suggestions. For this time, we are presenting the antioxidant activity as a percentage of inhibition. In our future works, we will certainly keep this in mind and will present the antioxidant activity expressed as a concentration as suggested.

Point 3: In addition, another control with Spirulina alone should be included to check its antioxidant activity individually.

Response 3: The nutritional composition and antioxidant activity of Spirulina powder alone has been presented in Table 1 of the revised script. Other tables have been re-numbered, accordingly.

Point 4: Review the bibliography by inserting DOIs

Response 4: The bibliography has been revised by inserting DOIs.

Round 2

Reviewer 3 Report

The authors have made the required revisions. I believe this work can be published.